# Mitochondrial Dysfunction and Inflammaging in Heart Failure: Novel Roles of CYP-Derived Epoxylipids

**DOI:** 10.3390/cells9071565

**Published:** 2020-06-27

**Authors:** Hedieh Keshavarz-Bahaghighat, Ahmed M. Darwesh, Deanna K. Sosnowski, John M. Seubert

**Affiliations:** 1Faculty of Pharmacy and Pharmaceutical Sciences, University of Alberta, Edmonton, AB T6G 2E1, Canada; hkeshava@ualberta.ca (H.K.-B.); darweshe@ualberta.ca (A.M.D.); dksosnow@ualberta.ca (D.K.S.); 2Department of Pharmacology, Faculty of Medicine and Dentistry, University of Alberta, Edmonton, AB T6G 2E1, Canada; 3Faculty of Pharmacy and Pharmaceutical Sciences, University of Alberta 2020-M Katz Group Centre for Pharmacy and Health Research 11361-87 Avenue, Edmonton, AB T6G 2E1, Canada

**Keywords:** aging, inflammasome, heart failure, mitochondria, N-3 and N-6 polyunsaturated fatty acids, epoxylipids

## Abstract

Age-associated changes leading to a decline in cardiac structure and function contribute to the increased susceptibility and incidence of cardiovascular diseases (CVD) in elderly individuals. Indeed, age is considered a risk factor for heart failure and serves as an important predictor for poor prognosis in elderly individuals. Effects stemming from chronic, low-grade inflammation, inflammaging, are considered important determinants in cardiac health; however, our understanding of the mechanisms involved remains unresolved. A steady decline in mitochondrial function is recognized as an important biological consequence found in the aging heart which contributes to the development of heart failure. Dysfunctional mitochondria contribute to increased cellular stress and an innate immune response by activating the NLRP-3 inflammasomes, which have a role in inflammaging and age-related CVD pathogenesis. Emerging evidence suggests a protective role for CYP450 epoxygenase metabolites of N-3 and N-6 polyunsaturated fatty acids (PUFA), epoxylipids, which modulate various aspects of the immune system and protect mitochondria. In this article, we provide insight into the potential roles N-3 and N-6 PUFA have modulating mitochondria, inflammaging and heart failure.

## 1. Introduction

Aging is a key determinant of cardiovascular health, as evidenced from an exponential increase in the prevalence of cardiovascular disease (CVD) in the geriatric population [1]. Aging hearts can be characterized by overall decreased function, reduced cardiac reserve capacity, structural remodeling and electrical dysfunction [2,3]. The extent and duration of exposure to extrinsic risk factors linked to development of CVD, including hypertension, diabetes and smoking increases with time, thereby contributing to increased cardiovascular morbidity and mortality in aged individuals [4,5]. In addition, naturally occurring biological aging events lead to the slow progressive deterioration in cardiac structure and function in the absence of systemic risk factors [6]. In essence, the age-associated extrinsic and intrinsic changes converge to accelerate a deterioration of cardiac function and structure contributing to the increased susceptibility and incidence of CVD in elderly.

Heart failure (HF) is a growing healthcare burden and the leading cause of hospitalization and readmissions [7]. HF is a clinical syndrome attributed to impaired ventricular filling or ejection of blood that can result from any abnormality in cardiac function and structure leading to an inability to adequately meet the hemodynamic requirements of the body at rest or with stress [8,9,10]. Heart failure is associated left ventricular (LV) dysfunction ranging from patients with normal LV structure and ejection fraction (EF) to those with severe LV dilation and reduced EF [10]. Based on EF, the guidelines classify HF into three categories: patients with preserved LVEF (≥ 50%, HFpEF), patients with mid-range EF (41–49%, HFmEF) and patients with reduced EF (≤ 40%, HFrEF) [9]. Epidemiological studies from patients with HF have reported that the proportion of patients with HFpEF is exceeding the number of those with HFrEF over recent years [11,12,13]. While HFpEF has a complex and heterogenous pathophysiology, ventricular diastolic dysfunction is considered the major outcome of the disease. The ventricular diastolic dysfunction is manifested as impaired diastolic relaxation accompanied by increased diastolic stiffness leading to abnormalities in ventricular filling dynamics [14]. Notably, these cardiac anomalies are aggravated by exposure to stress such as tachycardia, hypertension and even during exercise as EF does not increase proportionally with stress leading to impairment of systolic function [15,16,17,18]. Accumulating experimental and translational evidence demonstrated development of HFpEF cannot be attributed to a single cause; however, several factors contribute to the pathology [19]. For instance, vascular stiffness, systemic inflammation, as well as the presence of comorbid conditions such as diabetes mellitus and pulmonary hypertension, among others, have been implicated in the development of the disease [20]. The prevalence of HFpEF increases with age and is more prominent in women [21,22]. There are several age-related alterations in cardiovascular function and structure, such as atrial stiffening, vascular fibrosis and thickening, LV hypertrophy, and insufficient mitochondrial energy production. These distinctive changes may partly explain why elderly individuals are more likely to develop HFpEF [23,24]. However, because of inconsistent diagnostic criteria, limited understanding of pathogenesis, insufficient available data from human tissues, and lack of appropriate animal models, our understanding about the cellular and molecular mechanisms involved in HFpEF pathophysiology remains limited [25,26].

Regardless of LVEF, patients with HF have increased risk of mortality compared to those without HF [25]. The chief manifestations of HF are dyspnea, fatigue and exercise intolerance resulting from cardiac structural and functional abnormalities that lead to elevated intra-cardiac pressure or a reduced cardiac output [27,28]. Epidemiological studies show a growing prevalence of HF in the elderly, with people 65 years old and older constituting over 80% of the patients [29,30]. While aging itself does not cause HF, evolving evidence suggests that the direct effects of myocardial aging contribute to the development and progression [31]. In patients with HF, age has been determined to be an independent factor associated with poor prognosis and higher risk of cardiovascular events and mortality [32,33]. Importantly, a marked increase in aging populations is occurring globally where individuals over 65 are beginning to account for larger percentages of the total populace [34]. Given the dramatic growth in the elderly population, age-related HF represents one of the greatest challenges confronting the health care system.

Chronic inflammation is a key component of aging and aging-related pathologies associated with increased risk of cardiovascular morbidity and mortality [35]. “Inflammaging” refers to chronic low-grade inflammation resulting from long term physiologic activation of the innate immune system in the absence of overt infections [36]. Potential mechanisms of inflammaging include genetic susceptibility, cellular senescence, impaired autophagy, changes to microbiota composition, oxidative stress and dysfunctional mitochondria [29,37,38]. Chronic inflammation contributes to the decline in cardiac function, increased size of cardiomyocytes and myocardial fibrosis, furthering the development and progression of HF [39,40].

Studies in experimental animal models and human hearts suggest that mitochondria play a central role in the aging process and abnormalities in mitochondrial function and structure are considered major drivers of age-associated cardiac dysfunction [3,40,41]. In healthy myocardium, mitochondria provide up to 90% of energy demand of the beating heart by mediating electron transportation to generate ATP on a beat-to-beat basis [42]. Mismatch between ATP supply and demand attributed to mitochondrial dysfunction has been historically considered as the primary mechanism linking mitochondria to cardiovascular diseases [43,44,45]. However, the role of mitochondria is now increasingly recognized to reach far beyond a failed powerhouse [46,47]. Defective mitochondrial reactive oxygen species (ROS) handling has emerged as a central factor in pathogenesis of a wide variety of cardiovascular dysfunctions, including cardiac aging and HF [48]. Moreover, mitochondria are emerging as key players in regulation of innate immunity. Studies on mitochondrial-mediated regulation of inflammasomes and sterile inflammation in CVD have expanded increasingly in recent years [49,50,51]. Enhancing our understanding of inflammatory pathways interwoven with mitochondrial events in the process of cardiac aging remains important, notably toward developing more promising therapeutics for age-associated pathologies, including HF.

## 2. NLRP3 Inflammasomes in Chronic Inflammation and Heart Failure

A chronic low-grade inflammatory state involving the sustained activation of innate immune system is a key characteristic of cardiac aging [52]. The innate immune system comprises a non-specific arm of the host-defense representing an initial defensive response against invading pathogens and plays a central role in activating the adaptive immune response [53,54,55]. The innate immune system is activated through the recognition of pathogen associated molecular patterns (PAMPs) or damage associated molecular patterns (DAMPs) by specific receptors called pattern recognition receptors (PRRs), which are present in the extracellular and endosomal compartment [56]. While PAMPs are derived from microorganisms and initiate immune response to infections, DAMPs stem from host cells and are released in response to cellular stress or tissue damage [57,58]. The engagement of PPRs initiates signaling pathways that lead to release of pro-inflammatory cytokines, migration of additional innate immune cells and activation of the adaptive immune system mediated by B and T cells [59,60,61]. An important component of the pathway involves inflammasomes, which are intracellular multiprotein complexes activated in response to a wide variety of intrinsic and extrinsic danger signals. [62]. To date, the NOD-like receptor (NLR) family pyrin domain containing 3 (NLRP-3) inflammasome has been increasingly recognized to play a major role during aging processes [63]. The inflammasomes are comprised of a sensor molecule, NLRP-3 with a pyrin domain (PYD), and the adaptor protein, apoptosis-associated speck-like protein (ASC), which harbors pro-caspase-1. Upon activation, NLRP-3 interacts with ASC via PYD to form NLRP3-ASC-pro-caspase-1 complex, also known as NLRP-3 inflammasome [64,65]. Subsequently, caspase-1 becomes activated and proceeds to cleave pro-IL-1β and pro-IL-18 into mature IL-1β and IL-18, promoting an inflammatory response [66]. Activated caspase-1 also cleaves gasdermin D (GSDMD) and results in the formation of GSDMD-N terminal domain (GSDMD-N) [67]. The GSDMD-N triggers pyroptosis by translocating into the plasma membrane and inducing membrane pore formation, resulting in cell swelling, release of IL-1β and Il-18, and eventual lysis [68,69,70].

Accumulating evidence indicates a role for NLRP-3 inflammasomes in CVD and aging, contributing to the chronic activation of inflammatory pathways [63,64]. Experimental manipulation of the innate immune system by over-activating the NLRP-3 inflammasome induces a global age-related inflammatory response in various organs [71]. In this study, NLRP-3 was shown to be the key regulator in age-related increases in IL-1β and caspase-1 activity in the murine central nervous system (CNS), which was associated with the age-related reduction in cognitive function and motor performance [71]. Genetic ablation of NLRP-3 inflammasome in mice entails metabolic and systemic effects with an improvement in cardiovascular health, such as improved glucose tolerance, the regulation of dyslipidemia and metabolic pathways resulting in enhanced longevity [72]. The improved longevity and health induced by inhibition of NLRP-3 could be partially explained by an increased autophagic response observed in the mice that results in elimination of misfolded proteins in cardiac tissue [72]. Elevated levels of pro-inflammatory cytokines secondary to NLRP-3 inflammasome activation, such as IL-1β, are often detected in elderly individuals correlating with age-related CVD [38,73]. Indeed, the formation of NLRP-3 inflammasome promotes collagen production and IL-1β activation leading to adverse cardiac remodeling and caspase-1 mediated cell death [74,75]. Mature IL-1β promotes the LV expression of pro-fibrotic tumor growth factor β (TGF-β), excessive extracellular matrix (ECM) accumulation and collagen I deposition [76,77]. The subsequent development of myocardial hypertrophy and LV fibrosis are hallmarks of age-related reduced LV elasticity and diastolic dysfunction [78,79].

A consequence of chronic low-grade inflammation is the accumulation of senescent cardiomyocytes resulting in compromised cardiac function, augmented cell death and myocardial dysfunction [39]. In essence, persistent activation of inflammatory pathways, such as NLRP-3 inflammasomes, coupled with the effects of cardiac aging, set the stage for development of HF [40,66,80,81]. Accordingly, in the context of HF, increased levels of NLRP-3 inflammasomes and inflammatory mediators induced by pressure overload in murine hearts have shown to be associated with increased myocardial fibrosis, LV hypertrophy and impaired cardiac function, contributing to progression of the disease. For example, experimental evidence in rodents using Triptolide, a traditional Chinese medicine used for rheumatoid arthritis, demonstrated protective effects by the downregulation of NLRP-3/TGF-β pro-fibrotic axis and decreased expression of pro-inflammatory cytokines like IL-1β and IL-18 [82]. Human samples from stressed and injured myocardium demonstrate the presence of elevated levels of NLRP-3 inflammasome-containing immune cells, which potentially contribute to the worsening of HF [74]. While evidence is limited, experimental animal studies suggest inflammasome activation impacts the pathogenesis of HF. Data demonstrating the chronic activation of NLRP-3 is associated with the progressive decline in systolic dysfunction and reduced contractility, which was prevented by blockade of NLRP-3 and IL-1β [83]. Similarly, studies using the NLRP-3 inflammasome inhibitor, MCC950, resulted in slowing the development of HF in both models of myocardial infarction (MI) and pressure overload by suppressing IL-1β release [84]. Together, the current data reflect correlative associations and do not demonstrate causality; however, the continued presence of increased NLRP-3 inflammasome during aging and in HF suggest the importance of this pathway and the need for more research.

## 3. Aging Mitochondria Contribute to NLRP-3 Inflammasome Activation

Given the immense energetic cost of cardiac electrical and mechanical function and the limited capacity for energy storage, the heart mainly relies on mitochondria as a steady energy supply [44,85]. Approximately 95% of cardiac ATP is produced by mitochondria through oxidative phosphorylation. However, the importance of cardiac mitochondrial health and function is now increasingly recognized to reach beyond ATP synthesis [86]. Mitochondria play a central role in a myriad of cellular processes, such as oxidative stress homeostasis, biosynthetic pathways, signaling and programmed cell death [87]. It has long been appreciated that aging is accompanied by a decline in mitochondrial function and quality contributing to a wide variety of age-related diseases, including HF [52,88]. Aged cardiomyocytes show extensive mitochondrial abnormalities, including enlarged structure, loss of cristae, reduction in ATP synthesis, impaired dynamics and increased ROS production [40,79,86,89]. Historically, numerous studies have proposed the detrimental alterations in mitochondrial function and structure play a central role in myocardial hypertrophy, fibrosis and consequently transition to HF [90,91]. The role mitochondria have in activating NLRP-3 inflammasomes has led to new concepts for their involvement in aging and age-related diseases and mitochondria-induced inflammasome activation has become central to theories of cardiac aging [29,92,93].

## 4. Mitochondria and Oxidative Stress Theory of Aging

Dysregulated ROS production and impaired antioxidant defense have been implicated in a host of cardiovascular dysfunctions, including cardiac aging and HF [94]. Age-associated oxidative stress has been demonstrated in clinical studies, evidenced by depleted glutathione, impaired superoxide dismutase (SOD) and an increase in malondialdehyde levels in elderly individuals [95,96,97]. Excessive production of ROS triggers numerous adverse effects leading to cell dysfunction, lipid peroxidation and DNA mutagenesis ultimately resulting in irreversible cell damage and death [98]. Conversely, overexpression of antioxidant molecules, including mitochondrial thioredoxin (Trx) and catalase, has been suggested to extend the lifespan in animal models [99,100]. The overexpression of human Trx in mice protected murine bone morrow cells from ultraviolet C (UVC)-induced oxidative stress and improved the telomerase activity associated with increased maximum lifespan [99]. Elevated mitochondrial catalase (MCAT) activity in mouse hearts attenuated the severity of age-induced cardiomyopathy and arthrosclerosis reducing oxidative damage to cardiac DNA resulting in a longer median lifespan [100]. The occurrence of increased oxidative stress in aging is thought to be a driver of NLRP-3 inflammasome activation [101]. Several underlying novel pathways regulating age-associated oxidative stress have been elucidated, among which mitochondrial ROS generation is of particular importance in the setting of inflammasomes activation [38,51,66].

Mitochondrial respiratory chain serves as a major source of ROS production where leakage of single electrons are transferred to molecular oxygen forming superoxide anions [102]. Due to the proximity to the electron transport chain, mitochondrial DNA (mtDNA) is highly susceptible to ROS-mediated damage, leading to further mitochondrial dysfunction [103]. The vicious cycle between mitochondrial damage and the further overproduction of ROS causes the dysregulation of various cellular pathways, including inflammation and apoptosis, and eventually resulting in cardiac functional decline [39,104]. Interestingly, the leakage of oxidized mtDNA into the cytosol activates the NLRP-3 inflammasome in macrophages exposed to lipopolysaccharide (LPS) and ATP, where depletion of NLRP-3 and ASC attenuated the release of mtDNA [105]. Accordingly, cells lacking mtDNA are unable to secrete IL-1β in response to NLRP-3-activating stimuli, corroborating the active role of mtDNA in NLRP-3 inflammasome activation. More importantly, the oxidized form of mtDNA and not the normal form was shown to bind to and activate NLRP-3 inflammasomes [106]. However, the exact role of mtDNA in NLRP-3 inflammasome activation remains unknown [107]. NLRP-3 inflammasomes require a priming step usually stimulated by inflammatory cytokines, PAMPs or DAMPs, upregulating the expression of inflammasome components. This is followed by the activation step resulting in the inflammasome complex assembly and subsequent activation of caspase-1. A major challenge is understanding the mechanisms involved in the abrupt transition from priming to activation-induced NLRP-3 activators [108]. In a recent study by Karin and colleagues, they reveal the priming and activation of NLRP-3 inflammasome is coupled through the induction of mtDNA synthesis. The newly replicated mtDNA co-precipitates and co-localizes with NLRP-3 inflammasomes in response to NLRP-3 activators. These newly activated mtDNA molecules are also highly sensitive to oxidative stress, which further activates NLRP-3 inflammasomes [107]. This study sheds new light linking mitochondria to both the priming and activating stage of the NLRP-3 inflammasome.

Age-related accumulation of free cholesterol crystals is shown to serve as a pro-inflammatory trigger, activating NLRP-3 inflammasomes and the release of IL-1β [109]. Elevated cellular cholesterol levels impair mitochondrial membrane potential and respiratory function, resulting in increased levels of mtDNA in the cytosol [110,111]. Additionally, impaired mitochondrial structure and the release of mtDNA into the cytosol results in engagement of absent in melanoma 2 (AIM 2) inflammasome in a cholesterol-dependent manner resulting in exaggerated release of IL-1β [111]. Both pharmacological and genetic perturbation of cholesterol trafficking to the endoplasmic reticulum (ER) in macrophages inhibited NLRP-3 inflammasome activation and reduced the secretion of caspase-1 and IL-1β suggesting a significant role for cholesterol in inflammasome assembly and activation [112].

Based on the oxidative stress hypothesis for aging, both increased ROS levels and a decline in efficiency of the antioxidant system can contribute to the progressive degeneration in cardiac function and structure [113]. Mitochondrial thioredoxin system (Trx 2), localized to the mitochondrial matrix, is a free radical scavenger providing the primary defense against mitochondrial ROS. Trx is highly expressed in metabolically active tissues, such as cardiac cells, where it has a major role protecting cells against damage [114]. Thioredoxin interacting protein (Txnip) has been identified as a tumor suppressor protein with a primary role of inhibiting the antioxidant activity of Trx via direct interaction [115]. Upon stress conditions, Txnip is shuttled into mitochondria and inhibits Trx-2 leading to increased ROS production and leakage [116]. While Txnip was initially characterized as a key regulator in cellular redox signaling pathways, evidence suggests the function of Txnip goes beyond classical redox biology.

The age-dependent upregulation of Txnip correlates with lower Trx-2 expression levels, which together lead to the accumulation of ROS, increased oxidative stress and the perturbation of cellular redox equilibrium [114,117,118]. The increased Txnip expression has been documented in isolated primary T cells from elderly patients (> 55 years old) compared to young (20–25 years old) individuals [117]. Further studies in murine models of diabetic nephropathy and from human diabetic patients have revealed increased ROS production results in Txnip mediated activation of the NLRP-3 inflammasome cascade [119,120]. Downregulation of Txnip in THP-1 macrophages suppressed caspase-1 and secretion of IL-1β in response to inflammasome activators, indicating the Txnip/NLRP-3 axis is critical for the inflammatory response. Moreover, intraperitoneal injection of MSU, an inflammasome activator, in Txnip-deficient mice resulted in significantly lower neutrophil influx and IL-1β production [121]. Moreover, inhibiting the NLRP-3/Txnip axis in H9c2 cardiomyocytes suppressed doxorubicin-induced cardiac senescence and cell damage [122]. While our current understanding of the impact altered Txnip levels has toward adverse cardiac effects in aged individuals is limited, evidence suggests it may disrupt the redox status and activate the NLRP-3 inflammasome cascade.

## 5. Impaired Mitophagy and Mitochondrial Dynamics in Cardiac Aging and Heart Failure

Mitochondria are dynamic organelles constantly undergoing fission and fusion events in response to energy demand and cellular stress. The balanced fission and fusion events under basal conditions are responsible for maintaining mitochondrial morphology and metabolism [86]. Key proteins regulating fission include dynamin-related protein 1 (Drp-1) and fission protein 1 (Fis1), while mitofusin 1 and 2 (Mfn-1 and 2) and optic atrophy 1 (OPA1) are involved in mitochondrial fusion in mammals. Altered expression or activation of mitochondrial dynamic proteins have been implicated in the pathogenesis of cardiac diseases [42]. Cardiac specific ablation of Drp-1 gene in mice inhibits mitochondrial fission, resulting in mitochondrial enlargement, increased mitochondrial permeability transition pore (MPTP) opening, apoptosis, and ultimately, lethal dilated cardiomyopathy (DCM) [123]. Interrupting mitochondrial fusion with the deletion of Mfn-1 and Mfn-2 genes in mice also leads to progressive and lethal DCM, primarily due to disrupted mitochondrial structure and respiratory chain function [124]. Evidence suggests that the imbalance between mitochondrial fission and fusion during aging has a role in age-related CVD via compromising mitochondrial integrity [40,125,126]. Reduced fission and/or increased fusion have been shown to be associated with elongated, hyper-fused mitochondria in aged tissues [127,128,129]. For example, changes to the relative expression levels of Mfn-2 and Drp-1 in skeletal muscle of aged mice is associated with more elongated mitochondria [127]. Similarly, mitochondria from aged C. elegans showed significantly enlarged and swollen ultrastructure accompanied by decreased oxygen consumption, increased carbonylated protein and decreased mitochondrial SOD activity [129]. While an elongated morphology is associated with the accumulation of dysfunctional mitochondria in aged tissues, promoting Drp-1 mediated fission in midlife in Drosophila improves mitochondrial respiratory function and structure, prolongs lifespan and delays age-related pathologies [130]. The concomitant interruption of fission and fusion processes can accelerate mitochondrial senescence and result in the accumulation of dysfunctional mitochondria, contributing to the development of HF [104].

HF patients have aberrant mitochondrial dynamics and altered homeostasis [131,132]; as such, understanding the pathophysiology associated with dysfunctional fission/fusion processes in age-related CVD is important. Moreover, considering the importance of inflammasome signaling in the pathogenesis of age-associated HF, deciphering the role of altered mitochondrial dynamics with inflammasome activation is significant. While the mechanistic details are limited, hyper-fused mitochondria in mouse bone marrow macrophages, caused by knockdown of Drp-1, are capable of triggering NLRP-3 inflammasome assembly and activation of caspase-1 and IL-1β [133]. Moreover, the chemical induction of mitochondrial fission by carbonyl cyanide m-chlorophenyl hydrazine-attenuated inflammasome activation in the macrophages [133]. Conversely, ablation of Mfn-2 in murine macrophages, significantly reduces the activation of IL-1β following RNA virus exposure [134]. Together, while studies on the role of mitochondrial dynamics in HF have been very limited, emerging evidence suggests that the impaired mitochondrial fission/fusion balance are etiologically involved in promoting NLRP-3 inflammasome activation and contribute to the disease progression.

Mitophagy is a highly selective autophagic process of the degradation of damaged mitochondria necessary for maintaining cellular homeostasis, and is closely associated with mitochondrial dynamics [135]. Overtime, or as a consequence of stress, when mitochondria sustain damage too severe to overcome, they undergo mitophagic removal [136]. Diminished mitophagy results in cellular accumulation of dysfunctional mitochondria, reducing overall quality, which may contribute to the aging process [126]. There are multiple proposed mechanisms for the removal of damaged mitochondria, such as the serine/threonine kinase PTEN-inducible kinase 1 (PINK1)/E3 ubiquitin ligase (Parkin) pathway [137]. Upon mitochondrial damage and membrane depolarization, PINK1 accumulates on the outer membrane of mitochondria and mediates phosphorylation and subsequent activation of Parkin leading to ubiquitination of mitochondrial proteins, including Mfn-2 [135]. The recruitment of mitochondrial proteins promotes the interaction of damaged mitochondria to LC3-positive phagosomes for degradation in lysosomes. Any impairment of these processes can lead to mitochondrial dysfunction and cell death [138,139].

There is growing evidence from animal models and clinical data suggesting that dysfunctional mitophagy results in the accumulation of dysfunctional mitochondria and augments myocardial cell death and necrosis, thereby accelerating the progress toward cardiomyopathy and HF [139,140,141,142]. Both the expression and activity of PINK-1 were significantly reduced in mice with heart failure 14 days following transverse aortic constriction (TAC). Increasing cardiac mitophagy was observed in PINK-1 protected mice in a chronic TAC-induced HF model resulting in improved mitochondrial function [142]. Furthermore, studies in C. elegans and Drosophila demonstrate enhancing mitophagy promotes a protective mechanism against mitochondrial stress and results in the extension of lifespan [143,144,145]. Defective mitophagy characterized by the accumulation of damaged mitochondria was observed in accelerated aging in Werner syndrome (WS) in both human fibroblasts and C. elegans. Interestingly, treatment with Urolithin A, a mitophagy specific inducer, improved pharyngeal pumping of the worms and extended their lifespan [143]. Accordingly, despite not fully understood, impaired mitophagy has been critically linked to aberrant inflammatory responses, including NLRP-3 inflammasome activation [146]. Parkin deficiency enhances mitochondrial membrane potential loss, augments mitochondrial ROS production and mtDNA release leading to elevated activation of IL-1β and caspase-1, in response to NLRP-3 agonists [147]. Conversely, upregulated Parkin expression and enhanced mitophagy inhibits NLRP-3 inflammasome assembly and activation of downstream signaling molecules promoting cell survival [148].

Although many of these studies lack the mechanistic explanation, it can be hypothesized that insufficient mitophagy leads to the accumulation of dysfunctional mitochondria, excessive production of ROS and oxidized mtDNA, which in turn, triggers inflammasome activation [144,149,150]. Together, the current data suggest an important intersection between mitochondrial dynamics, mitophagy and inflammasome activation in the progression and development of age-associated CVD.

## 6. Macrophages and Chronic Inflammation

Aging is an inevitable part of life, posing the largest risk factor for all myocardial diseases [151]. Higher levels of fibrosis, cardiomyocyte loss and cellular hypertrophy are morphological hallmarks of aged hearts contributing to worsened outcomes of almost all cardiac pathologies, including HF [152]. Accumulating evidence suggests aging is associated with a general remodeling of the immune system leading to an “autoimmune” sterile inflammatory response that can contribute to cardiac structural changes [153,154]. Multiple mechanisms have been proposed to explain how chronic low-grade inflammation impacts the process of cardiac aging, ranging from mitochondrial dysfunction and metabolic disorders to dysregulation of immune cells [155,156].

Macrophages are integral components of the innate immune system and are critical in the homeostatic maintenance of the myocardium under steady-state conditions and after tissue injury [157]. In addition to their classical roles in host defense, macrophages are involved in cardiac remodeling, healing and clearing senescent cells contributing to tissue homeostasis [158]. The predominant belief was tissue macrophages were simply derived from circulating monocytes, however, it is now well-established that cardiac resident macrophages are derived from embryonic hematopoietic progenitors, independent from infiltrating monocytes [159]. Despite the existence of a heterogeneous macrophage population in the mammalian heart, macrophages are broadly subdivided into pro-inflammatory M1 and anti-inflammatory M2 phenotypes [160]. Although both M1 and M2 macrophages can be divided into various subcategories, tissue macrophages are traditionally categorized into anti-inflammatory M2 population and recruited infiltrating monocyte-derived macrophages are generally categorized as a M1 inflammatory group [161]. While direct evidence of the role of macrophages in inflammaging and age-associated diseases is scarce, a shift in macrophage populations in aged hearts in favor of M1 phenotype suggests involvement in age-related cardiac deterioration [162]. Using genetic fate mapping and parabiotic approaches, Molawi et al. indicated that the number of resident macrophages declines in aged hearts and infiltrating monocytes progressively contribute to the cardiac macrophage population [163]. In addition to the decreased ratio of M2/M1 macrophages, the self-renewal capacity of cardiac macrophages is dampened with physiological aging, which may contribute to the vulnerability and worsened prognosis of an aged heart following injury [164,165]. The quantification of the total number of cardiac macrophages by staining the proliferation antigen phospho-histone 3 (PH3) showed a decline in the number of cardiac macrophages in 30-week-old mice compared to their 4-week-old counterparts. Interestingly, cardiac macrophages from 15–25-week-old mice demonstrated upregulation of pro-fibrotic genes, such as matrix metallopeptidase 9 (MMP-9), TGF-β1 and fibroblast growth factor receptor 1 (FGFR-1) [165]. Accordingly, an age-associated progressive loss of M2 macrophages is linked to a prolonged inflammatory status accelerating cardiac aging [166].

Aging is associated with increased accumulation and circulation of DAMPs that are hypothesized to play a major role in the age-associated low-grade inflammation [167]. Accumulating evidence suggests macrophages are primary responders in age-related sterile inflammation and persistent activation of macrophages by age-associated DAMPs is one of the main mechanisms in inflammaging and tissue dysfunction [168]. Mechanistically, mitochondrial dysfunction and increased oxidative stress occurring with age and M1 polarization may result in hyper-activation of NLRP-3 inflammasome and IL-1β release contributing to age-related pathologies [169,170]. Telomere attrition is a hallmark of the aging process which can be reversed by the action of telomerase enzyme. Telomerase-deficient macrophages showed abnormal mitochondrial function and ultrastructure accompanied with increased mROS production resulting in activation of NLRP-3 inflammasome. Further relating the impaired mitochondria to inflammasome activation, treatment with mito-TEMPO, a specific scavenger of mROS, significantly attenuated the enhanced activation of caspase-1 and IL-β in telomerase-deficient macrophages [169]. Although the endogenous mechanisms linking macrophages to aging and longevity is unknown, inhibiting the macrophage-derived inflammatory cytokine cascade successfully improved lifespan in drosophila [171]. Evidence suggests the downregulation of NLRP-3 inflammasomes in macrophages maintains immune homeostasis in aged-tissues leading to pro-longevity effects [172]. Together, the data suggest M1 macrophage associated inflammasomes drive adverse age-associated low-grade inflammation while M2 macrophages provide anti-inflammatory benefits.

## 7. N-3 and N-6 Polyunsaturated Fatty Acids (PUFAs)

Long-chain polyunsaturated fatty acids (PUFA) are essential fatty acids obtained from dietary sources which are required for cellular organelles, such as phospholipid membranes, and serve as precursors to numerous bioactive lipid mediators [173]. Linoleic acid (LA) is the primary source of N-6 PUFA, which is converted to arachidonic acid (AA), while α-linolenic acid (ALA) is considered the main N-3 PUFA precursor [174]. Inside the body, the conversion of ALA and LA to their corresponding downstream metabolites happens through multiple elongation and desaturation steps [175]. ALA can be converted into eicosapentaenoic acid (EPA) which can be further metabolized to yield docosahexaenoic acid (DHA); however, the conversion is very limited in humans [176]. Since the N-3 and N-6 PUFAs compete for the same metabolic pathways, the dietary N-3:N-6 PUFA ratio plays a critical role in normal tissue function and development [174]. A typical western diet provides higher levels of LA resulting in predominant generation of LA-derived eicosanoids, which results in an unbalanced ratio correlating with the etiology of many diseases, including CVD [175,177].

Emerging evidence demonstrates the epoxy, hydroxyl and diol metabolites derived from N-3 and N-6 PUFA metabolism have important properties. The metabolism of N-3 and N-6 PUFA occurs primarily through three enzymatic systems, cyclooxygenases (COX), lipoxygenases (LOX) and cytochrome P450 (CYP) enzymes, into a plethora of bioactive metabolites. Various members of the CYP superfamily are capable of metabolizing N-3 and N-6 PUFAs into bioactive lipid mediators (Figure 1).

In the cardiovascular system, CYP2J and CYP2C isozymes are major epoxygenases responsible for converting AA, a N-6 PUFA, into four regioisomeric epoxyyeicosatrienoic acids (5,6-, 8,9-, 11,12-, and 14,15-EET) by olefin epoxidation [178,179]. As well, converting the N-3 PUFAs, EPA into 5 regioisomeric epoxyeicosatetraenoic acids (5,6-, 8,9-, 11,12-, 14,15-, 17,18-EEQ) and DHA into 6 regioisomeric epoxydocosapentaenoic acids (4,5-, 7,8-, 10,11-, 13,14-, 16,17-, 19,20-EDP) [180]. In the heart, EETs act as the key lipid mediators, regulating cellular mechanisms, including mitochondrial quality control, apoptosis, and inflammatory pathways [181,182,183]. Although little is known about the exact mechanisms of the cardioprotective effects of N-3 PUFAs, recent evidence demonstrates that CYP-derived epoxy metabolites possess anti-inflammatory and cardioprotective effects [184,185]. Most N-3 and N-6 epoxylipids, including EETs, EDP and EEQ, have a short half-life and are rapidly metabolized to their corresponding less active diol metabolites by the enzyme soluble epoxide hydrolase (sEH) [186,187].

Over the past decade, experimental studies have demonstrated EETs mediate a myriad of cellular and metabolic pathways, which are cardioprotective toward several pathologies including HF [187,188,189,190]. For instance, Cao et al. reported that using (*S*)-2-(11-(nonyloxy) undec-8(*Z*)-enamido) succinic acid (NUDSA), an EET agonist, in a murine model of myocardial infarction (MI) is associated with improved systolic dysfunction, decreased myocardial fibrosis and limited remodeling in post-infarcted HF [191]. Similarly, the administration of an orally active EET mimetic in hypertensive rats exposed to ischemia-reperfusion-induced HF, reduced cardiac associated mortality, provided better cardiac function, reduced pulmonary edema, reduced myocardial fibrosis and decreased macrophage infiltration. Moreover, EET mimetic treatment increased the activity of hem-oxygenase 1 (HO-1) following ischemia-reperfusion injury and attenuated the progression of HF [192]. In a murine model of HF, treatment with sEH inhibitor, adamantan-3-(5-(2-(2-ethylethoxy) ethoxy) pentyl) urea (AEPU) to limit epoxylipid metabolism, prevented pressure-overload-induced cardiac hypertrophy and decreased susceptibility to ventricular arrhythmias. Furthermore, treatment with AEPU decreased the translocation of NF-κB from the cytosol into the nucleus in mouse neonatal cardiomyocytes subjected to Angiotensin II (Ang II)-induced hypertension and hypertrophy [193]. Moreover, transgenic mice with cardiomyocyte overexpression of CYP2J2 subjected to pressure-overload or long-term infusion of isoproterenol demonstrated reduced hypertrophy and arrhythmogenic events [194]. Evidence suggest that EETs play a major cardioprotective role in HF by decreasing secretion of pro-fibrotic factors leading to reduced remodeling [195]. Mice with cardiac overexpression of CYP2J2 showed improved cardiac function, decreased myocardial hypertrophy and fibrosis resulting in amelioration of cardiac remodeling. Further defining the cardioprotective role of CYP2J2, neonatal cardiomyocytes from mice with cardiac overexpression of CYP2J2 showed decreased level of cardiac remodeling proteins, such as collagen type I, and TGF-β in response to Ang II compared to their WT counterparts [195]. Although the exact mechanism of the cardioprotective effects of EETs remains unknown, these studies established evidence that EETs have promising therapeutic effects for improving cardiac outcomes in HF.

The evidence suggesting N-3 PUFAs have cardioprotective effects against pathological hypertrophy and HF is controversial [187]. Experimental animal studies suggest there are beneficial effects of N-3 PUFAs toward CVD. For example, in a mouse model of HF, increased myocardial EPA and DHA levels following dietary-supplementation-attenuated LV chamber dilation against pressure-overload-induced cardiomyopathy providing a proof-of-concept. Moreover, following TAC, mice fed with EPA and DHA showed improved mitochondrial function documented by preservation of citrate synthase and medium chain acyl-CoA dehydrogenase (MCAD) activity [196]. However, differences in the clinical literature from several prospective observational studies and large-scale clinical trials testing the protective effects of N-3 PUFAs have had mixed results [197]. The “GISSI-HF” trial demonstrated N-3 PUFA supplementation was associated with reduced HF-related hospital admissions and mortality in patients with reduced ejection fraction [198]. Conversely, a randomized double-blind trial by the “Alpha-Omega Trial Group” concluded that there was no significant benefit for N-3 PUFA toward cardiovascular events post-MI [199]. These differences may be partially attributed to study design, such as the inclusion of populations with a high baseline intake of N-3 PUFA and differing doses of EPA and DHA, yet no studies to date investigate the role of the CYP-derived epoxy metabolites.

Both preclinical and clinical studies have furnished a wealth of evidence in support of cardioprotective effects of epoxy fatty acids [187,190,196,200]. However, their short half-life limits their therapeutic use and clinical application requiring new strategies to improve their pharmacokinetics [187,201]. The gene encoding sEH, *Ephx2*, is the primary enzyme metabolizing PUFA epoxides resulting in the formation of respective diol metabolites via the addition of a water molecule [182,183]. Novel pharmacological approaches that selectively inhibit sEH have evolved as clinical tools in various cardiovascular diseases, including hypertension, cerebral ischemia, cardiac ischemia, cardiac hypertrophy, myocardial infarction and atherosclerosis [202,203,204,205,206,207]. Pharmacological sEH inhibition is demonstrated to improve both LV diastolic and systolic function and attenuate myocardial remodeling in established HF [188,208,209]. Treatment with *cis*-4-[4-(3-adamantan-1-yl-ureido) cyclohexyloxy] benzoic acid (c-AUCB)-attenuated increased systolic and diastolic LV cavity diameter following ischemia/reperfusion injury in rats. Interestingly, the combined administration of EET mimetic and c-AUCB amplified the cardioprotective effects of the single therapy and significantly decreased the MI-induced chamber dilation accompanied with improved systolic function [188]. Chronic inhibition of sEH substantially attenuated lung congestion and albuminuria parallel to preserved cardiac function and structure, acknowledging sEH as a therapeutic target for the treatment of HF associated with chronic kidney disease [210,211]. Despite considerable research on the role of sEH, significant gaps remain at many levels in the understanding of mechanisms involved in beneficial effects of sEH inhibition in the setting of cardiovascular diseases.

CYP-dependent oxidation of LA results in the production of epoxyoctadecanoic acids (EpOME), which are rapidly metabolized by sEH to their corresponding diol metabolites, dihydroxyoctadecanoic acids (DiHOME) [187]. Evidence from animal and in vitro studies has suggested DiHOMEs are potent cytotoxic metabolites [212]. Increased levels of cardiac DiHOMEs are thought to be associated with deteriorated myocardial electrical activity, altered ion channel kinetics, depressed LV function and impaired mitochondrial respiration [213,214,215,216]. The cardiotoxic effects of DiHOMEs remained evident when cardiac specific over-expression of CYP2J2 failed to improve cardiac functional recovery following ischemia reperfusion, attributed to age-related accumulation of DiHOMEs in the heart [217]. Indeed, the deleterious myocardial effects of DiHOMEs counter the cardioprotective effects of increased epoxylipids [218]. These data suggest that the cardioprotection of sEH inhibition could be mediated, at least in part, by inhibiting the production of cardiotoxic DiHOMEs. However, further investigation into cardiac effects of DiHOMEs is warranted.

## 8. Anti-Inflammatory Effects of N-3 and N-6 PUFA-Derived Epoxylipids

Several experimental, clinical and epidemiological studies reported that the cardioprotective effects of the epoxylipids derived from N-3 and N-6 PUFAs against various cardiac pathologies are attributed to their immunomodulatory properties. The anti-inflammatory properties of N-3 and N-6 PUFAs as well as their corresponding epoxylipids have been demonstrated in both cardiac and extra-cardiac tissues [219,220]. For example, EETs ameliorated inflammatory responses in human endothelial cells subjected to TNF-α-stimulated inflammation, by decreasing the expression of cytokine-induced adhesion molecule, vascular cell adhesion molecule–1 (VCAM-1) and preventing leucocyte adhesion attributed to inhibition of nuclear transcription factor-kappa B (NF-kB) activity. Supportive investigations determining the effects of EETs in vascular inflammation demonstrated intra-arterial perfusion of 11,12-EET decreased endothelial VCAM-1 expression and prevented the adhesion of mononuclear cells in mice following injection of TNF-α [219]. Moreover, the increase in EET levels associated with the cardiac specific overexpression of CYP2J2-attenuated atrial fibrosis and inflammatory responses in a mouse model of atrial fibrillation induced by constriction of the abdominal aorta [221]. Treatment with EET analogs limited myocardial fibrosis and macrophage infiltration, decelerating the progression of MI-induced HF in rats subjected to ligation of left anterior descending (LAD) coronary artery [192]. While the exact mechanisms remain unknown, current data provides compelling evidence that anti-inflammatory properties of EETs play a major role in their cardioprotective effects.

Emerging evidence indicates a beneficial immunomodulatory role for N-3 PUFA involves limiting mitochondrial dysfunction and subsequent NLRP3 inflammasome activation in hearts subjected to ischemia-reperfusion injury. Hearts perfused with DHA or 19,20-EDP demonstrated preserved mitochondrial quality, as evidenced by the reduced mitochondria translocation of Drp-1, Txnip, preserved OPA-1 expression and better mitochondrial function [204,222]. In another study, TNF-α-stimulated primary human retinal microvascular endothelial cells treated with exogenous EETs or EDPs in combination with a sEH inhibitor, 12-(3-adamantane-1-yl-ureido)-dodecanoic acid (AUDA) resulted in inhibited leucocyte adherence and attenuated retinal vascular inflammation. AUDA prevented the metabolism of both endogenous and exogenous EET and EDP and attenuated TNF-α-induced VCAM-1 expression [223]. Similarly, when obese mice subjected to high-fat diet were treated with EPA and DHA, a robust anti-inflammatory response was observed involving attenuation of NF-kB activation and TNF-α pro-inflammatory cascade. Interestingly, in the same study, genetic deletion of G-protein coupled receptor 120 (GPR-120) completely abrogated the effects of EPA and DHA supplementation on insulin sensitivity and glucose infusion rate in mice subjected to a high-fat diet [224]. In a randomized, double blind cross-over controlled study, DHA and EPA supplementation in trained endurance athletes for ten weeks, significantly lowered values of muscle inflammatory markers, including IL-6 and IL-1β, paralleled with reduced muscle damage indices [225]. In another study on a murine model of hypertension treated with angiotensin (Ang)-II, co-administration of sEH inhibitor with EPA and DHA effectively decreased Ang-II-dependent hypertension and attenuated release of renal inflammatory markers, such as MCP-1 and prostaglandins [226]. The emerging evidence is suggesting that N-3 PUFAs and their metabolites have important anti-inflammatory properties suggesting novel therapeutic approaches in CVD.

There is a growing body of evidence demonstrating both genetic and pharmacological inhibition of sEH abrogate the uncontrolled sterile inflammation in different settings [187,227,228]. For instance, Yang et al. showed the sEH inhibitor, *trans*-4-{4-[3-(4-trifluoromethoxyphenyl)-ureido]-cyclohexyloxy}-benzoic acid (t-TUCB), increased concentrations of the anti-inflammatory EETs and simultaneously decreased the concentrations of type 2 T helper cells (Th2), cytokines (IL-4, IL-5) and chemokines as well as total cell and eosinophil numbers in the lung lavage of a murine ovalbumin (OVA) model of asthma [229]. The sEH inhibitor 12-(3-adamantan-1-yl-ureido)-dodecanoic acid butyl ester (AUDA-BE) and 1-adamantan-3-(5-(2-(2-ethylethoxy)ethoxy)pentyl)urea (compound 950), increased EET levels, reduced the concentration of the pro-inflammatory cytokines, such as TNF-α, IL-6 and MCP-5, accelerated inflammatory resolution and prevented mortality in a mouse model of septic shock [230]. The inhibition of sEH significantly ameliorated inflammatory responses, attenuated neutrophil infiltration and decreased the expression of pro-inflammatory cytokines, such as TNF-α and IFN-γ in a mouse model of inflammatory bowel disease treated with piroxicam [231]. In a rat model of ischemic stroke induced by middle cerebral artery occlusion, the administration of a pharmacological inhibitor of sEH decreased the mRNA level of IL-1β and IL-6 and modulated microglia polarization toward decreased M1 and increased M2, resulting in reduced infarct size and improved behavior outcomes [232]. Furthermore, incubation of primary murine macrophages subjected to LPS with novel dual COX-2/sEH inhibitor-attenuated ROS production, reduced the level of caspase-1 and IL-1β and inhibited NLRP3 activation [233]. Although the balance of evidence suggests substantial benefits of sEH inhibitors with respect to CVDs, the exact cardioprotective mechanism of sEH inhibition remains controversial and an area of active investigation.

LA is metabolized to EpOMEs by CYP epoxygenases and further to DiHOMEs by sEH [187]. DiHOMEs are characterized as pro-inflammatory mediators in pain contributing to the pathophysiology of the disease [234]. Further, 12,13-DiHOMEs promote monocyte infiltration and expression of pro-inflammatory cytokines resulting in inflammatory immunologic responses [235]. Increased epoxide hydrolase expression and 12,13-DiHOME concentrations are associated with an increased probability of developing childhood allergies associated with inflammatory properties [236]. There is limited information regarding the adverse impact of DiHOMEs toward cardiac function and aging. However, evidence suggests the myocardial accumulation of 12, 13-DiHOME correlates with LPS-induced cytotoxicity and a decline in mitochondrial function and cell survival [212]. The 12,13-DiHOME-augmented release of ROS and mitochondrial dysfunction in hearts exposed to ischemia-reperfusion is associated with diminished myocardial functional recovery [213]. Together, sEH inhibition results in increased levels of epoxylipids while decreasing DiHOME production potentially altering cardiac inflammatory responses.

## 9. Mitochondria: Effects of N-3 and N-6 PUFA-Derived Epoxylipids

Mitochondria play a fundamental role in cardiac aging by regulating a plethora of age-associated cardiac changes, particularly the age-induced pro-inflammatory status [237]. Therefore, targeting mitochondrial dysfunction in the aging myocardium is an unmet need holding significant promise for age-related cardiac diseases. Although the exact mechanisms of how epoxylipids regulate cardiac function are not fully understood, accumulating data suggest mitochondria-targeted effects are an important component of their cardioprotective properties [203,212,238]. Numerous in vivo and ex vivo studies demonstrate sEH inhibition or treatment with epoxylipids improve LV functional recovery in murine hearts following ischemia-reperfusion injury protecting mitochondrial function and ultrastructure [181,206,225,238,239,240,241]. Previously, we reported hearts perfused with UA-8 (13-(3-propylureido) tridec-8-enoic acid) a synthetic dual-action compound possessing EET mimetic and sEH inhibitory properties, improved post-ischemic contractile function and reduced infarct size following ischemia-reperfusion injury. These cardioprotective effects were attributed to the ability of UA-8 to prevent the collapse of mitochondrial function and limit the loss of mitochondrial membrane potential, resulting in preserved heart function [242]. The inhibition of endogenous EET production by a selective epoxygenase inhibitor, MS-PPOH, resulted in the disruption of mitochondrial ATP generation, increased ROS production, mitochondrial depolarization and mitochondrial fragmentation in cultured neonatal hippocampal astrocytes [243]. Both treatment with exogenous EET and CYP2J2 overexpression suppressed ROS production and increased expression of catalase, as well as cytosolic and mitochondrial superoxide dismutase leading to improved viability in human pulmonary artery endothelial cells subjected to anoxia/reoxygenation [244]. The administration of an EET agonist effectively ameliorated obesity-induced cardiomyopathy by improving mitochondrial function and energy metabolism in cardiac tissues leading to enhanced tolerance to glucose challenge, associated with increased cardiac expression of PGC-1α, a key regulator in mitochondrial biogenesis [245]. Moreover, in the same study, EET-treated mice showed increased level of manganese superoxide dismutase (MnSOD) and Mfn-2 in adipose tissues, contributing to balancing the mitochondrial function and redox status [245]. Ablation of the sEH gene in mice decreased the degradation of EETs leading to a significant increase in Mfn-1, HO-1 and the cytochrome c oxidase subunit I (COX-I) level in adipose tissue compared to WT control mice, further revealing a key role for EETs in regulating mitochondrial integrity and function [246]. The sEH deletion also maintained both mitochondrial function and ATP generation in isolated murine cardiac fibers subjected to LAD in both young and aged mice, contributing to sustained systolic and diastolic function. Moreover, both young and aged sEH null mice had preserved mitochondrial ultrastructure following myocardial infarction (MI) characterized by improved cristae density and organization [206].

Although the effects of N-3 PUFAs on mitochondria are less extensively studied and characterized, they are increasingly recognized to protect the heart by preserving mitochondrial function [247,248]. A DHA-rich diet significantly improved the ROS-induced MPTP opening in interfibrillar mitochondria and decreased mitochondrial membrane viscosity associated with a modest attenuation of LV dysfunction in rats with HF [249]. Both EPA and DHA exhibited an up-regulatory effect on the expression of Mfn-2 resulting in a significant recovery of mitochondrial network architecture and morphology accompanied by increased ATP production in steatotic HepG2 cells incubated with oleate and palmitate [250]. The anti-apoptotic and pro-survival effects of N-3 PUFAs by shifting the cell death pathway toward survival have been also reported [183,244]. DHA attenuated apoptosis, evidenced by increased Bcl-2 and decreased Bax and cleaved caspase-3 via upregulating OPA-1 and ameliorating mitochondrial fragmentation following subarachnoid hemorrhage in rats subjected to ligation of the carotid artery [251].

Sirtuin 3 (Sirt-3) is a nicotinamide adenine dinucleotide (NAD)-dependent histone deacetylase found predominately in mitochondria which has been identified as a key mediator in age-related cardiovascular physiology, regulating mitochondrial oxidative stress via deacetylating MnSOD [252,253]. Cardiomyocytes lacking Sirt-3 show age-dependent mitochondrial swelling and accelerated signs of cardiac aging, including myocardial hypertrophy and accumulated fibrotic tissue [254]. Interestingly, while the cardiac expression of sEH is significantly increased in cardiac aging, the genetic deletion of sEH attenuated the age-related decrease in Sirt-3 activity in female mice [255]. The effect was associated with higher levels of active mitochondrial MnSOD resulting in better overall cardiac function, suggesting the preservation of mitochondrial integrity in aged mice [255]. This data fosters an important body of research on the underlying mechanisms involved in the effects of epoxylipids on mitochondrial redox apparatus in cardiac aging and age-related pathogenesis.

The dysregulation of mitochondrial dynamics and mitophagy found in aged hearts is associated with an accumulation of damaged mitochondria and subsequent activation of inflammatory responses [126]. The suppression of sEH demonstrated to increase expression of Mfn-1 associated with improved cardiac mitochondrial function and biogenesis, increased ATP production, ameliorated cardiac inflammation and consequently protected the heart from metabolic syndrome [256]. EETs also maintained the increased expression of Mfn-2 and MnSOD in obesity-induced cardiomyopathy shedding more light on their role in maintaining mitochondrial homeostasis [245]. While the mechanisms remain unknown, there appears to be a role for epoxylipids in regulating mitochondrial function in aged hearts.

## 10. Sex Differences and N-3 and N-6 Polyunsaturated Fatty Acids

Biological aging is associated with slowly progressive deterioration in cardiac function and structure, predisposing elderly adults to cardiovascular diseases [257]. There is growing evidence that significant sex differences exist in presentation, progression and treatment responses as individuals age, resulting in different clinical outcomes between men and women and highlighting the importance and necessity of considering sex differences in aging studies [258,259,260]. Males present greater LV mass and chamber dimensions with increased susceptibility toward developing eccentric LV remodeling, systolic dysfunction and DCM at younger ages [261]. However, upon aging, increased myocardial wall thickness has been reported to occur in females, which may be accompanied by concentric LV chamber remodeling and diastolic dysfunction, predisposing females to HF [258,260]. Furthermore, there are several cardiovascular risk factors exclusively experienced by women, including menopause, early menarche, preeclampsia, and pregnancy [262]. The higher incidence of CVD in men compared to women prior to menopause suggests sex hormones play key role in development [263]. However, emerging research has indicated sex hormones alone are insufficient to fully explain the variations in cardiac outcomes between men and women [264]. Interestingly, the data pertaining to biological sex-differences show distinct disparities in male and female mitochondrial function and morphology [265]. Female mitochondria are more differentiated with increased cristae density and protein content resulting in lower levels of free radical production and higher efficiency [265,266,267]. In view of the fact that mitochondria are strongly associated with age-related cardiac differences, comparable mitochondrial sex-differences appear to be a promising theory to explain sex-specific differences in cardiac aging. Sexual dimorphism has been documented in protein expression and regulation of sEH activity in both cardiac and extra-cardiac tissues [255,268,269,270]. While sEH inhibition did not have discernable effects on female mouse systolic blood pressure (SBP), genetic deletion of sEH efficiently normalized SBP in male mice [270]. Sexual disparity in regulating sEH activity becomes more pronounced in aged tissues, as aging is associated with a significant increase in sEH in male animals. This raises new questions on the mechanisms associated with sex-specific responses to sEH-based interventions [255,268].

## 11. Conclusions

Our understanding of the progression of cardiac aging is largely exclusive to a collection of intriguing yet unconnected data sets bridging the age-associated inflammasome activation and age-related CVD. Identifying novel approaches to modulate aged-induced immune responses will provide insights into the development of effective therapeutics with limited adverse effects. In the current review, we briefly discussed the immunomodulatory properties of PUFA-based epoxy fatty acids and sEH inhibitors. Several mechanisms contribute to the immunomodulatory effects of epoxylipids, from which maintaining mitochondrial structure and function is of great importance in the platform of aging and age-related HF. Taken together, while our understanding of the role of epoxylipids in modulating the immune system remains limited, several findings have highlighted their diverse cardioprotective activities and provided ideas toward developing novel therapeutic approaches (Figure 2).

## Figures and Tables

**Figure 1 cells-09-01565-f001:**
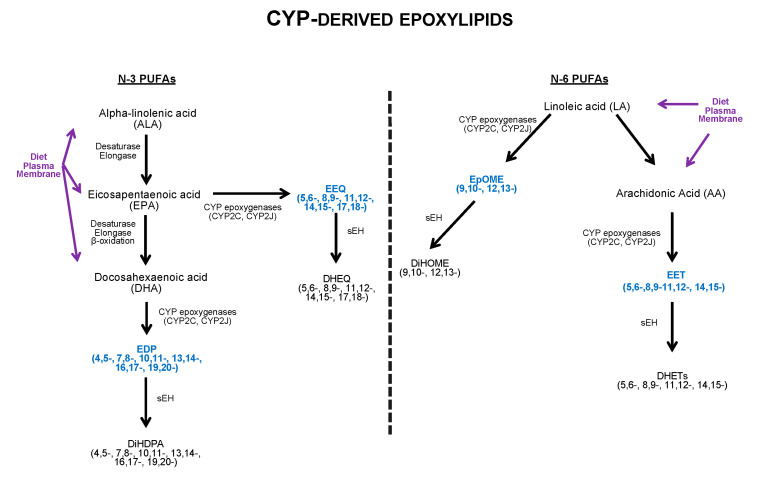
Overview of metabolic pathway of CYP-derived epoxylipids. LA, AA, ALA, EPA, and DHA are essential dietary fatty acids and are found in membrane phospholipids. AA: arachidonic acid; ALA: α-linolenic acid; CYP: cytochrome P450. DiHOME: dihydroxyoctadecenoic acid; DHA: docosahexaenoic acid; DiHDPA: dihydroxydocosapentaneoic acid; DHEQ: dihydroxyeicosatetraenoic acid; DHET: dihydroxyeicosatrienoic acid; EET: epoxyeicosatrienoic acid; EDP: epoxydocosapentaenoic acid; EEQ: epoxyeicosatetraenoic acid; EpOME: epoxyoctadecamonoenic acid; LA: linoleic acid; PUFA: poly unsaturated fatty acid, sEH: soluble epoxide hydrolase.

**Figure 2 cells-09-01565-f002:**
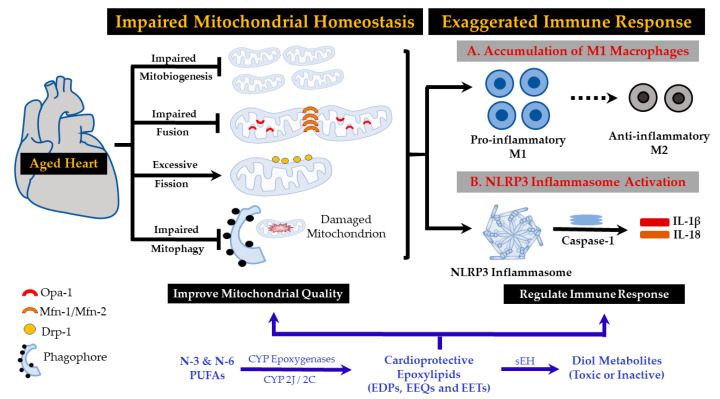
Schematic diagram of the potential modulatory effects of cytochrome P450 (CYP)-derived epoxylipids against cardiac inflammaging. Cardiac senescence is an intrinsic process accompanied by a general decline in mitochondrial function and impaired mitochondrial homeostasis, as evidenced by reduced mitochondrial biogenesis, dysregulated fusion, exaggerated fission and suppressed mitophagy. These changes result in the accumulation of damaged mitochondria, which triggers an innate immune response characterized by cardiac infiltration of pro-inflammatory M1 macrophages and assembly of the NLRP3 inflammasomes. N-3 and N-6 PUFAs can be metabolized by CYP isoenzymes to their corresponding epoxylipids, which (i) maintain/improve mitochondrial integrity and function and (ii) attenuate NLRP3 inflammasome activation, suggesting a proof-of-concept for beneficial effects against inflammaging and age-related cardiac pathologies. DRP-1: dynamin-related protein 1, EET: epoxyeicosatrienoic acid, EDP: epoxydocosapentaenoic acid, EEQ: epoxyeicosatetraenoic acid, IL: interleukin, MFN: mitofusin, NLRP3: NOD-like receptor family, pyrin domain containing 3, OPA-1: optic atrophy 1, PUFA: poly unsaturated fatty acid, sEH: soluble epoxide hydrolase.

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
