# Peer review of "Mitochondrial Dysfunction and Inflammaging in Heart Failure: Novel Roles of CYP-Derived Epoxylipids"

_cells, 2020, doi:10.3390/cells9071565_

Round 1

Reviewer 1 Report

Keshavarz-Bahaghighat et al provide a narrative review on mechanisms underlying cardiac aging, focusing on the aspect of chronic inflammation and mitochondrial dysfunction specifically in heart failure.

Major Points:

The definition for heart failure (HF) used by the authors “HF is defined by an inefficient contractile function of the heart resulting in an inability to adequately meet the hemodynamic requirements of the body” is at odds with current definitions. Current definitions are broader and are more focused on symptoms and signs that can be caused both by heart failure with reduced ejection fraction (HFrEF) and heart failure with preserved ejection fraction (HFpEF), e.g. PMID’s 27206819 or 23741058. Heart failure is the central topic of the review article, HFpEF occurs with age, is responsible for roughly 50% of heart failure cases, but is not even mentioned. Therefore, the review thoroughly fails to meet the self-proclaimed aim defined in lines 76 – 78 of the manuscript. The omission of HFrEF is not understandable, and fundamentally questions competence and diligence invested by the authors also in all other aspects of the manuscript. This is illustrated by the fact that molecular mechanisms induced by exercise training, an important aspect of the therapy of patients with both HFrEF and HFpEF included in current guidelines (e.g. PMID 27206819), are not mentioned in the manuscript.

Reviewer 2 Report

This is a well-written review highlighting the roles of CYP-derived epoxylipids in heart failure. The roles of mitochondrial failure and inflammaging have particularly been highlighted and previous literature has been extensively cited. However, at places, it was felt that the review is a bit monotonous and repeatedly talks about mitochondrial dysfunction, ROS, mitophagy, and inflammasome activation. Moreover, many studies are discussed only superficially. It might be useful to discuss in-depth for some of the studies what parameters were studied to denote dysfunctional mitochondria. Additionally, I would also suggest bringing in lipids and epoxylipids from quite early on e.g. in the abstract and briefly in the introduction. In the latter part of the review, inflammasomes have not been mentioned at all – bringing in lipids, inflammasomes, and HF together will add much-needed significance to this article.

 Further suggestions below:

  1. Line 92: Inflammasomes do not become signalling platforms but it is the initial assembly of different components (NLR, ASC, and casp-1) that gives rise to inflammasomes. Inflammasomes are signalling platforms the moment they are assembled.
  2. Related to the above: The structure of this paragraph needs to improve: the authors describe the assembly of the inflammasome complex two times within 2-3 sentences in this paragraph. This should be corrected. Also, it should be described here that the pyroptosis induced by this complex is gasdermin D-dependent and appropriate articles should be cited.
  3. Page 110: LV needs to be expanded.
  4. Line 118: Is NLRP3 activation a cause or a consequence of injured myocardium in this study?
  5. Line 143: Describe here the new theories of cardiac ageing.
  6. In mitochondria and oxidative theory of ageing, there is no mention of AIM2 inflammasome activation by mtDNA. The authors have included the mtDNA synthesis study by the Karin lab. However, newer studies have revealed that loss of cholesterol 25-hydroxylase activates the AIM2 inflammasome. Similarly, ER cholesterol levels are known to regulate the NLRP3 inflammasome. Inclusion of discussion on these aspects is very important, even if not directly relevant, to a review on epoxylipids. How would results from these studies impact cardiac function and ageing? Again, papers related to lipids should be discussed from quite early on in the article.
  7. There is also a disconnect in the writing – inflammasomes and mitochondria have been extensively discussed however almost nothing has been mentioned in the context of the epoxylipids. Involvement of inflammasomes and mitochondria wherever relevant in the later section of the article would streamline the review.
  8. Abstract first sentence: Please correct: ‘decline in cardiac structure’.
  9. Abstract, Line 30: Please correct: In this article, we provide insight into the potential roles N-3 and N-6 PUFA have in modulating mitochondria’.
  10. Line 138: do you mean an enlarged structure?

Reviewer 3 Report

Keshavarz-Bahaghighat et al. write a nicely written and thourough review regarding 'inflammaging' and mitochondrial dysfunction in the development and progression of heart failure. In addition, this review does a nice job in covering n-6 and n-3 PUFAs and there metabolites in the regulation of inflammation, oxidative stress and mitochondrial regulation. Lastly, the authors include two nice figures that link their review topics; these figures add to the overall quality of the review. Outside of minor spell checks I really have no concerns regarding this review paper.  

Round 2

Reviewer 2 Report

Most of my concerns have been taken care of by the authors. It would have helped the authors if they had revised the abstract further but I understand.